# Approaching Saint Bernard's Sermons on the "Song of Songs" through the *Book of Odes* (*Shijing*): A Confluence of Medieval Theology and Chinese Culture

**Yanbo Zheng**

Religious Sciences, École Pratique des Hautes Études (EPHE), Paris Sciences Lettres University (PSL), 75014 Paris, France; yanbo.zheng@etu.ephe.psl.eu

**Abstract:** This paper aims to decode medieval theology from the vantage point of ancient Chinese poetry, employing a cross-textual methodology that encourages a fusion of horizons. It highlights Saint Bernard's profound and influential theological exegesis of the "Song of Songs", particularly his comparison of the divine–human relationship to the conjugal bond. The present study posits that readers from Chinese culture can gain access to Saint Bernard's mystical theology through the sentiment of love, as portrayed in the *Book of Odes* (*Shijing*). Initially addressing love as a core human sentiment, this study progresses by juxtaposing the representations of love in the *Book of Odes* with those in the "Song of Songs". This comparative analysis culminates in an exploration of Saint Bernard's theological perspectives, illuminated through these analogous depictions of love. The results affirm that engaging with Saint Bernard's discourse on love via the *Book of Odes* is not only feasible but also instrumental in dispelling widespread misconceptions.

**Keywords:** medieval mystical theology; *Book of Odes* (*Shijing*); "Song of Songs"; Saint Bernard of Clairvaux

## 1. Introduction

Love, an enduring and universal theme, has captivated philosophers, theologians, writers, poets, and artists across all epochs, who have endeavored to articulate their experiences and interpretations of it. Descartes, widely recognized for his assertion "Cogito, ergo sum", probes the foundation of self-existence by questioning the certainty of external realities versus the indubitable presence of a doubting self. Jean-Luc Marion, a contemporary French philosopher, critiques Descartes' emphasis on thought, suggesting that he overlooks the primordial role of "love". Marion argues that the inability to envisage being loved constitutes a profound erosion of selfhood, reducing it to a mere mechanistic entity devoid of the relational capacity observed even among animals. He posits that openness to love precedes rational cognition, asserting that existence is sustained not merely through rational justification but also through the intrinsic value of love (Marion 2007, p. 26). This perspective reframes the ego not as an autonomous thinker but as an inherently relational one grounded in the act of love.

In contemporary society, the absence of love as a foundational principle risks reducing significant achievements to mere exercises in instrumental rationality. For instance, in activities such as marriage, child-rearing, participation in community or club engagements, caregiving to parents, visits to hospitals and nursing homes, and even participation in religious practices, if these activities are not underpinned by love, they may devolve into choices made purely out of instrumental rationality. Attempts to discourse on love frequently encounter interpretative ambiguity. Efforts to assign it a precise, rigorous definition are often futile due to the vast diversity in its objects and expressions, ranging from divine love and familial affection to friendship, spousal devotion, sexual partnership, altruism towards strangers, and fervor for ideals or concepts. Concepts such as duty,

rights, pleasure, benefits, and expectations further obscure our direct articulation of love. This complexity has spawned diverse terminologies to capture the nuances of related yet distinct emotional experiences. Reexamining and reflecting upon historical articulations and contemplations of love appears meaningful for contemporary society.

While love inherently constitutes a relational dynamic with another, its orientation frequently extends towards a transcendental realm, embodying a yearning that remains perpetually unfulfilled. Emmanuel Levinas elucidates this notion by suggesting the presence of luminosity "beyond the face, from what is not yet, from a future never future enough, more distant than the possible" (Levinas 1990, p. 285). A historical examination of the notion of love, particularly its transcendental attributes, reveals a rich tapestry of discourse among ancient scholars. This reflective inquiry sets the stage for a deeper exploration of the ancient conceptions of love, with an emphasis on the significant impact of the biblical narrative, especially as articulated through the "Song of Songs". This scriptural composition, devoted to the exploration of love between a man and a woman, beckons a wider spectrum of interpretations and understandings transcending its explicit narrative. The "Song of Songs" is unique in the biblical canon for centering its thematic essence on the romantic love between a man and a woman. The exegetical contributions of Saint Bernard of Clairvaux to this dialogue are notably profound. His series of interpretations not only garnered widespread recognition but also constituted some of the most celebrated and influential analyses during the Middle Ages, laying foundational principles for mystical theology.

The legacy of Saint Bernard's exegetical contributions to the "Song of Songs" can be evaluated from two critical dimensions. Initially, the broad influence exerted by the 13th-century seminal work, *Golden Legend*, served as a conduit for disseminating Saint Bernard's insights, notably impacting the Dominican order throughout the 14th century and subsequently permeating medieval Scholastic philosophy (Boureau 1993, pp. 85–86). Saint Thomas Aquinas, a pivotal theologian of the Middle Ages, was also influenced by Saint Bernard's exegesis of the "Song of Songs" (Bonino 2019, pp. 20–21). The interpretative endeavors of Saint Bernard concerning the "Song of Songs" had an indirect but significant influence on the philosophical frameworks of John Duns Scotus (Boulnois 1999, p. 166). Moreover, within the Cistercian order, Saint Bernard's profound influence ensured that his unfinished commentary on the "Song of Songs" found successors in Guillaume de Saint-Thierry and John de Forde, who endeavored to complete his interpretative vision. The implications of their contributions are profound, marking an indelible impact on the broader currents of Western Monasticism. This dual-faceted legacy highlights the enduring scholarly and spiritual resonance of Saint Bernard's work in the "Song of Songs" across diverse theological and philosophical landscapes.

When Chinese readers encounter Saint Bernard's *Sermons on the Song of Songs*, how is understanding possible amidst the collision of two distinct cultures? This question pertains to Gadamer's discussion on the "fusion of horizons".

> This is not to open the door to arbitrariness in interpretation but to reveal what always takes place. Understanding the word of tradition always requires that the reconstructed question be set within the openness of its questionableness—i.e., that it merges with the question that tradition is for us. If the "historical" question emerges by itself, this means that it no longer arises as a question. It results from the cessation of understanding—a detour in which we get stuck. Part of real understanding, however, is that we regain the concepts of a historical past in such a way that they also include our own comprehension of them. Above I called this "the fusion of horizons". (Gadamer 2004, p. 367)

Gadamer posits that interpreters and texts are shaped by 'effective history', framing understanding within a hermeneutic horizon. Prejudices from this history are foundational to interpretation, which can evolve through a 'fusion' with the studied work (Van Zoeren 1991, p. 5). Consequently, it is imperative to scrutinize the intrinsic textual significance of both the "Song of Songs" and *Shijing* prior to delving into the effectiveness of the fusion of horizons.

In *the Analects* of Confucius, *Ren* emerges as a multifaceted concept, appearing 109 times in *the Analects*, and is closely associated with love. Yet, romantic expressions are scarcely found in the Confucian texts, with the *Book of Odes* (*Shijing*, henceforth) standing as a notable exception. Both *Shijing* and the "Song of Songs" highlight the presence of the love poem at the inception of both Chinese and Hebrew civilizations, respectively. While comparisons between Confucian and Western metaphysical thoughts are frequent, the juxtaposition of the "Song of Songs" with *Shijing* is less common. This suggests that Confucianism alone cannot encapsulate Chinese culture entirely, which is a view critiqued by Marcel Granet, a French sinologist who argues against James Legge's view that Confucianism fully represents Chinese civilization (Granet 1982, p. 16). Noteworthy comparative work on these texts explores their civilizational origins and literary expressions (Gálik 1997). Both texts reveal intricate nuances and robust declarations of love, showcasing similarities in how love is experienced and expressed. The present study focuses on the authentic human sentiment of love delineated within these works, echoing Granet's call for a deeper analysis that transcends literary and symbolic interpretations to unveil the original essence of the poems in *Shijing* (Granet 1982, p. 17).

We observe that both the "Song of Songs" and *Shijing* are accompanied by a rich tradition of commentary. In the Confucian school of China, the earliest interpretation of *Shijing* was naturally conducted by the founder of the Confucian school, Confucius himself. According to the records of his students' observations and interactions with him, Confucius primarily focused on employing *Shijing* to elaborate on his views regarding ritual norms (Van Zoeren 1991, p. 43). A century or two after Confucius' death, Confucian teachers transformed *Shijing* into texts imbued with moral significance (Van Zoeren 1991, p. 52). They often extracted verses from their original context within *Shijing* and endowed them with new meanings. In the fourth century BC, Mencius further interpreted *Shijing* based on Confucius' life and teachings. He approached the book from a distinctly Confucian standpoint, from which he developed the Confucian philosophy of life and political ideals. Xunzi, a Confucian writer and philosopher who lived during the Warring States Era in China, also frequently cited *Shijing* to substantiate his ethical doctrines, though his aim was to introduce his legalistic ideas (Van Zoeren 1991, p. 76).

During the Han Dynasty, with the then emperor's support, Confucianism saw greater development, and interpretations of *Shijing* became increasingly rich. "New Text Confucianism was exemplified by the three schools represented by chairs in the Imperial Academy of the mid-second century. Each of these Three Schools (*Sanjia*), as they came to be known later, had its own text of the *Odes* and its own characteristic tradition of exegesis" (Van Zoeren 1991, p. 97). Thinkers during the Jin dynasty (266–420) and Northern and Southern dynasties (386–589) went further in imbuing *Shijing* with moral and educational meanings, mapping the positions of different poems within *Shijing* to the spatial structure of the world and even the universe—a perspective not verified by modern astronomy.

Kong Yingda (574–648), a distinguished Confucian scholar during the Tang Dynasty, offered an interpretive analysis of "The Ospreys Cry" *(Kwan-Ts'ü)* that diverged significantly from its original intent, opting instead for a political exegesis. He construed the poem as a representation of the conjugal dynamics between an emperor and his consort, extending this metaphor to encompass the rulers across various dynasties. Kong Yingda elucidated the intricate linkage between the emperor's emotional state and the broader realms of national tranquility and economic progression, thereby imbuing the text with a profound political dimension (Saussy 1993, p. 76).

Ouyang Xiu (1007–1072) of the Song Dynasty reflected on the traditions of *Shijing* commentary of the Han, Jin, Northern, and Southern Dynasties, opposing the overly allegorical interpretations of the Han Dynasty thinkers and advocating for a more direct reading of *Shijing* (Van Zoeren 1991, p. 167). He was arguably the first thinker since the Han and Tang Dynasties to candidly acknowledge *Shijing* as a collection of love poetry. For instance, his interpretation of "Of Fair Girls" (*Tsing nü*) appears sincere and straightforward:



The medieval interpretation as developed by Zheng Xuan and the Correct Significance grew out of that of the Mao Commentary and the Preface. The Preface states that the Ode was directed against Duke Xuan of Wei (r. 717-699 BCE), who, with his wife, was "guilty of licentious conduct". On this reading, the Ode described a beautiful and virtuous young woman, an appropriate mate and implicit reproach for the duke. The "corner of the Wall" in the first stanza was emblematic of the young woman's unassailable virtue (because the wall was high), and the red tube of the second was a writing brush with which the "female historians" (nüshi) of the Zhou court recorded the rules and activities of the harem. By so rebuking the duke and his consort, the poet demonstrated an appropriate emotional response to their conduct, and the Ode that inscribed that response could serve as an instrument of moral education. Ouyang dispensed with most of what he termed the "far-fetched" (yu) features of the medieval interpretation. For Ouyang, the Ode concerned a meeting between two lovers; the "red tube" of the second stanza was a love token, and the corner of the wall a trysting spot. Although he was able, by a deft interpretive move to be described below, to avoid challenging the central assumption of the tradition that the Odes represented morally paradigmatic attitudes, Ouyang nevertheless radically transformed the way that the Ode was read, producing a reading that was refreshingly simple and direct. It is for such reinterpretations that he is justly famous in the history of the reading of the Odes. (Van Zoeren 1991, p. 170)

Nonetheless, Ouyang Xiu's exegesis of *Shijing* remained embedded within a broader Lebensphilosophie. Subsequent Neo-Confucian scholars also critically addressed the propensity to employ *Shijing* for political interpretations. Cheng Yi (1033–1107), in particular, scrutinized *Shijing*'s efficacy in the refinement of individual character. He interpreted *Shijing* through the lens of the *Great Preface*, advocating for a methodological engagement with the text that transcends a mere literal comprehension. Cheng Yi encouraged readers to liberate themselves from subjective biases by engaging in *wanwei*, a meticulous and reflective exploration of the text, aiming to internalize its conveyed meanings deeply. While his approach was ostensibly pedagogical, Cheng Yi underscored the dynamic interplay between the reader and *Shijing*'s narrative, seeking to encapsulate the spectrum of emotions embedded within (Van Zoeren 1991, p. 209).

Zhu Xi (1130–1200), revered as one of the preeminent metaphysical thinkers in the annals of Confucian scholarship, navigated the interpretative landscape of *Shijing* with a discernible departure from elementary emotional expressions. He critiqued certain poems in *Shijing* as morally degenerate, which is a perspective steeped in ethical judgment. This stance, however, overlooks the fluidity and contextuality of moral standards across various epochs. Zhu Xi posited that Confucius intentionally incorporated these ostensibly debauched poems within *Shijing* as pedagogical counterexamples, which is a testament to the text's instructive underpinnings (Van Zoeren 1991, p. 229).

A brief examination of the interpretative traditions of *Shijing* reveals a tension between the traditions of exegesis and the affections expressed within *Shijing*'s text itself (Saussy 1993, p. 74). It is common practice to extricate verses from *Shijing* from their original contexts, imbuing them with new meanings. However, the Confucian school did not entirely forsake the affections found within *Shijing*. The founder of Confucianism, Confucius, did not hesitate to commend the beauty and sincerity of the sentiments within *Shijing*, as demonstrated by his statement: "In the [*Kwan-Ts'ü*], joy is found without frivolity; sorrow without detrimental effect" (Analects 3.20). In response to criticisms from other schools of thought, the Confucian scholars of the 2nd century BC refocused on the emotions within *Shijing* (Van Zoeren 1991, p. 54). The tradition of interpreting *Shijing*, particularly by Mencius, indeed discusses emotion, but it does so within the context of the relationship between emotion and *Zhi* (life's objectives) rather than as an unmediated, primal, unreflected sentiment (Van Zoeren 1991, pp. 112–13). The aim of this paper is to spotlight the unadulterated and unassigned genuine emotions within *Shijing*, attempting, through a

comparative analysis with the "Song of Songs", to uncover the reflections of universal emotional expressions inherent to humanity. If we accept the shared human nature across civilizations, we concur that engaging with *Shijing* facilitates Chinese readers in accessing the mystical theological context of Saint Bernard.

Prior to the reintroduction of Aristotle's works to Europe, Saint Bernard of Clairvaux emerged as a pivotal intellectual figure within the Latin world. His contributions were central to medieval spiritual thoughts, embodying both asceticism and mysticism. Notably, in Dante Alighieri's *Divine Comedy*, he is portrayed as the final guide through Paradise, and he is often celebrated as one of the last Church Fathers.

Saint Bernard's lifetime, spanning from the 11th to 12th centuries, coincided with the Crusades, a period marked by the emergence of the first cities in Western Europe, the zenith of Romanesque art, and the nascent stages of Gothic art. This era also witnessed the flourishing of vernacular literature, a resurgence in Latin classical texts and poetry, and the establishment of the foundational principles of Europe's first universities. Intellectual pursuits were predominantly driven by three movements: the Carthusian School, initiated by Guigues I le Chartreux; the Cistercian School, co-founded by Saint Bernard and his disciple Guillaume de Saint-Thierry; and the Victorine School, founded by Hugh of Saint Victor and Richard of Saint Victor (Gilson 1934, pp. 13–18).

Saint Bernard posited that the ultimate nature of intellect and transcendent love culminates in the superiority of love, which he argued is more conducive to the highest form of knowledge, or "unknowing". This stance is distinct from the later philosophy of "voluntarism". His exegetical work on the "Song of Songs"—*Sermons on the Song of Songs*—interprets this biblical book as a metaphor for mystical union, elevating human love to a divine level. His analysis diverges from the Platonist–Dionysian tradition of "ecstasy", offering a unique perspective on mystical experiences. Saint Bernard's primary emphasis was not on the uniqueness of his own experiences nor on portraying himself as superior to others. Rather, he sought to internalize his mystical experiences within the teachings of the Catholic Church and the traditions of the Bible.

## 2. The Similarities in the Description of Love between *Shijing* and the "Song of Songs"

*Shijing* is the beginning of ancient Chinese poetry, the earliest existing anthology of poems. It comprises a total of 305 poems from the 11th century BC to the 6th century BC. Formed before the establishment of the Confucian school, these poems describe the sentimental life of the Chinese people in ancient times. For reasons of space, it is not possible for us to compare these 305 poems one by one, but the author has selected eight of them as typical poems to be used for comparison with the "Song of Songs". It warrants emphasis that the arrangement of the poems discussed within this paper does not conform to the categorizations traditionally espoused by the Confucian tradition. This deviation assumes greater significance in light of the fact that certain scholars within ancient Confucianism interpreted the sequence within *Shijing* as underpinning a politico-cosmological framework. Given this perspective, maintaining fidelity to such a sequence is deemed less imperative in the context of this scholarly examination. We will find that there are many identical or similar sentiments about love in the two works.

The translations used in this paper are the ones of Arthur Waley's translation of *The Book of Songs* (Waley 1996). Nonetheless, Bernhard Karlgren's rendition of *Shijing* retains significant scholarly utility, notably through his meticulous preservation of the Chinese phonetic pronunciations for the titles of each poem within *Shijing*. As a result, despite my primary reliance on Arthur Waley's translations for analytical purposes, I have consistently included, in parentheses following each poem's title, the transliterated names as rendered by Bernhard Karlgren. This methodological approach facilitates the identification of specific texts within the corpus of *Shijing* across varying translation frameworks (Karlgren 1950). In an effort to circumvent the potential limitations associated with the exclusive use of one language to comprehend another, I have additionally referenced French versions of *Shijing*. The edition chosen for this purpose was translated by Pierre Vinclair (2019). This approach

enriches the analysis by introducing diverse linguistic perspectives in the text. The quotes from the "Song of Songs" in this paper are taken from the New American Bible version.

The first poem is called "Plop Fall the Plums (*Piao Yu Mei*)",

"Plop Fall the Plums
Plop fall the plums; but there are still seven.
Let those gentlemen that would court me
Come while it is lucky!
Plop fall the plums; there are still three.
Let any gentleman that would court me
Come before it is too late!
Plop fall the plums, in shallow baskets we lay them
Any gentleman who would court me
Had better speak while there is time".

The lady in the poem is encouraging her suitors to seize the opportunity without further delay or hesitation—a truly bold and progressive declaration. The poem directly shows the lady's inner thirst for love and that she wants to seek, urge, and call for love from males. When expressing her inner sentiments, the lady's tone of voice shows urgency and frankness without coyness. "Shedding is the plum-tree, its fruits are seven; . . .Shedding is the plum-tree, its fruits are three; . . .Shedding is the plum-tree, in a slanting basket I take them (the fruits)". By identifying herself with the ripening and falling plum, she expresses candidly that her youth is fading, and she wishes to cherish it by embracing love. So, when she expresses her feelings, she expresses them directly without beating around the bush so as to let her suiters know how ardent her desire for love is. The lady says directly to these men, "If any of you are fond of me, speak up quickly". We can see that the language is direct and forthright. In the French translation, the final sentence is rendered as "it is your turn to speak" (à vous de parler). This interpretation diverts the emphasis from temporal aspects to the act of speaking (parler), underscoring a particular embodiment associated with the mouth (Vinclair 2019, p. 50). In a similar vein, the "Song of Songs" begins with an unabashed expression of a lady's heart, "Let him kiss me with kisses of his mouth! More delightful is your love than wine! Your name spoken is a spreading perfume—that is why the maidens love you" (Song 1:2–3). The parallel between the forthright plea in *Piao Yu Mei* and the passionate yearning expressed in the "Song of Songs" underscores a timeless and universal longing for love. Just as the diminishing fruits of a plum tree, the lady's urgency reflects a keen awareness of the fleeting nature of youth and the importance of seizing the moment. This direct approach to expressing love, devoid of pretense and hesitation, highlights a boldness and clarity in communication that transcends cultures and epochs. The invocation of love in both poems serves as a reminder of the human desire for connection and intimacy, urging those who yearn to speak their heart's desire to do so without delay.

The second poem is called "Mad Boy (*Kiao T'ung*)", which goes as follows:

"That mad boy
Will not speak with me.
Yes, all because of you
I leave my rice untouched.
That mad boy
Will not eat with me.
Yes, it is all because of you
That I cannot take my rest".

The poem describes how a young lady, desiring a stroll and a picnic with her beau, feels disappointed. The inner sentiment described here is that of the lady's anxiety and resentment toward the object of her affection, which is due to the fact that her lover does not act in accordance with her expectations and that her emotional needs and expectations are

not met, thereby developing despondency. In the French translation, the translator employs "petit filou" to render *Kiao T'ung* (Vinclair 2019, p. 119), thereby diluting the connotations of condemnation inherent in "mad". The underlying significance of this choice merits further investigation. When lovers encounter frustration and disappointment within romantic relationships, the critical distinction lies in whether they attribute these feelings internally or externally. A narrow focus on the appellation "mad boy" may inadvertently overshadow the nuanced undertone of self-blame embedded within the anguish—evidently portrayed in the poem through her disturbed state, unable to find solace in eating or resting. A parallel disposition is articulated within the "Song of Songs", indicating a thematic resonance across these texts. The "Song of Songs" contains similar descriptions of inner activities, "tell me, you whom my heart loves, where you pasture your flock, where you give them rest at midday, Lest I be found wandering after the flocks of your companions" (Song 1:7). Both of these texts, in two different languages, express the frustration of the person who loves but cannot be requited. They express the initiative of the person who is unable to love and who, after being frustrated, turns to the pursuit of the object of his or her desire. This is clearly not a matter of cultural subtlety or exuberance but a reaction rooted in human nature. The essence of these verses lies in the raw depiction of longing and the turmoil of unreciprocated affection, painting a portrait of love's complexities that resonate deeply within the human psyche across different times and ages.

The third poem is called "Of Fair Girls (*Tsing Nü*)",

> "Of fair girls the loveliest
> Was to meet me at the comer of the Wall.
> But she hides and will not show herself;
> I scratch my head, pace up and down.
> Of fair girls the prettiest
> Gave me a red flute.
> The flush of that red flute
> Is pleasure at the girl's beauty.
> She has been in the pastures and brought for me rush-wool,
> Very beautiful and rare.
> It is not you that are beautiful;
> But you were given by a lovely girl".

This poem narrates from the perspective of a young male who is a little shy about giving a gift to the one he loves but feels great joy inside. The "Song of Songs" similarly describes the sincere sentiment of a lover who wants to put on the most beautiful object for his or her loved one: "We will make pendants of gold for you, and silver ornaments" (Song 1:11). In the face of the lover, it is natural for people, driven by voluntary generosity to give gifts to each other. This is not only to please each other but also as part of love's natural flow. It is not a transaction but an emotive tendency of human nature.

The fourth poem is known as "You with the Collar (*Tsi kin*)", a beloved classic that remains popular and relevant in today's Chinese society. Its full text is as follows:

> "Oh, you with the blue collar,
> On and on I think of you.
> Even though I do not go to you,
> You might surely send me news?
> Oh, you with the blue collar.
> Always and ever I long for you.
> Even though I do not go to you.
> You might surely sometimes come?
> Here by the wall-gate
> I pace to and fro.
> One day when I do not see you
> Is like three months".

We can see that one looks forward to his or her loved one's coming, and the "Song of Songs" also depicts such a mental activity, "My lover speaks; he says to me, Arise, my beloved, my beautiful one, and come!" (Song 2:10) and "Until the day breathes cool and the shadows lengthen, roam, my lover, Like a gazelle or a young stag upon the mountains of Bether" (Song 2:17). What these two texts describe is the lover's natural initiative and giving upon entering a romantic relationship, but also the desire for the loved one to be more proactive in demonstrating love for their partner.

The name of the fifth poem is "So Grand (*Kien Hi*)", which praises the body of the beloved one:

> "So grand, so tall
> He is just going to do the Wan dance;
> Yes, just at noon of day,
> In front of the palace, on a high place,
> A big man, so warlike,
> In the duke's yard he dances it.
> He is strong as a tiger.
> He holds chariot reins as though they were ribbons.
> Now in his left hand he holds the flute.
> In his right, the pheasant-plumes;
> Red is he, as though smeared with ochre.
> The duke hands him a goblet".
> "On the hills grows a hazel-tree;
> On the low ground the licorice.
> Of whom do I think?
> Of a fair lady from the West.
> That fair lady
> Is a lady from the West".

In this poem, the beloved man is tall and strong; he looks robust and flexible in dancing. When seeing his performance, the lady praises his beauty. Certainly, this poem represents an observer's commendation for a dancer, highlighting an evaluative moment transcending mere performance appreciation. Upon meticulous analysis, it is evident that the observer's reflections extend beyond the immediacy of the dance. Her considerations are not limited to the elegance of the dancer's movements; she explicitly recognizes the dancer's beauty and speculates about the dancer's place of origin. Although the dance scenario does not explicitly facilitate a direct emotional exchange between lovers, the emotional engagement of the female observer is undeniable. These emotions, in a comprehensive sense, may be classified as manifestations of love, propelled by the intrinsic instincts characteristic of human nature. In the English translation, the term "fair lady" is used to describe the dancer, which introduces a degree of ambiguity. In contrast, the French version opts for "belles personnes" (Vinclair 2019, p. 70), a choice that accentuates the poem's exploration of a broader human psychological state—one that recognizes the beloved's beauty in a gender-neutral manner. While the focus of this discussion is not on historical accuracy, it is imperative to acknowledge certain pervasive phenomena within human societies. Generally, dancers performing in public and receiving accolades from individuals of higher social standing tend to occupy a lower social echelon. On the contrary, women in antiquity who had the means to be spectators at such performances were likely of a comparatively elevated social status. Therefore, the poem embodies not merely an expression of genuine emotion but also a candor unobscured by the veneer of civilization. In the "Song of Songs", there are several descriptions of songs celebrating the figure of the beloved, "Ah, you are beautiful, my beloved, ah, you are beautiful; your eyes are doves! Ah, you are beautiful, my lover—yes, you are lovely. Our couch, too, is verdant" (Song 1:15–16). In *Kien Hi*, the red face of the beloved man attracts the lady. The "Song of Songs" contains precisely the exact same description, "My lover is radiant and ruddy; he stands out among thousands" (Song 5:10). There are more specific descriptions and metaphors about the beauty of the

beloved in the "Song of Songs", "Ah, you are beautiful, my beloved, ah, you are beautiful! Your eyes are doves behind your veil. Your hair is like a flock of goats streaming down the mountains of Gilead. Your teeth are like a flock of ewes to be shorn, which come up from the washing, All of them big with twins, none of them thin and barren. Your lips are like a scarlet strand; your mouth is lovely. Your cheek is like a half-pomegranate behind your veil. Your neck is like David's tower girt with battlements; A thousand bucklers hang upon it, all the shields of valiant men. Your breasts are like twin fawns, the young of a gazelle that browse among the lilies" (Song 4:1–5). It is not the intention of this paper to examine the meaning of the symbolism behind these different descriptions in terms of metaphor and analogy, and in the purest and most basic sense, we find that what is described in both *Shijing* and the "Song of Songs" is that the lover looks at his or her loved one and finds him or her to be extremely beautiful, and therefore praises him or her. This is a common occurrence in romantic relationships.

The sixth poem is so classic it is known as "The Ospreys Cry (*Kwan-Ts'ü*)", and it is as follows in its full text,

> "'Fair, fair', cry the ospreys
> On the island in the river.
> Lovely is this noble lady,
> Fit bride for our lord.
> In patches grows the water mallow;
> To left and right one must seek it.
> Shy was this noble lady;
> Day and night he sought her.
> Sought her and could not get her;
> Day and night he grieved.
> Long thoughts, oh, long unhappy thoughts.
> Now on his back, now tossing on to his side.
> In patches grows the water mallow;
> To left and right one must gather it.
> Shy is this noble lady;
> With great zither and little we hearten her.
> In patches grows the water mallow;
> To left and right one must choose it.
> Shy is this noble lady;
> With bells and drums we will gladden her".

The poem is an expression of a young man who pursues a lady of his choice, but he does not obtain a response from the lady. Therefore, the man cannot sleep at night due to his unrequited love. In this poem, the man projects his emotions of pining for and not being able to sleep on such images as trees and rivers. And in the "Song of Songs", the expression of having trouble sleeping is present because he misses his lover and cannot: "I was sleeping, but my heart kept vigil; I heard my lover knocking: Open to me, my sister, my beloved, my dove, my perfect one! For my head is wet with dew, my locks with the moisture of the night" (Song 5:2). The "Song of Songs" also uses images of the external environment to show inner sentiments. Both texts detail scenarios wherein individuals, due to intense affection, find themselves unable to sleep, lying awake with thoughts of their beloved. This depiction serves to underscore the universality of such sentimental experiences across different cultures. Furthermore, due to the intensity and ineffability of such sentiment, individuals are compelled to use external objects as metaphors to express these experiences of sentiment.

The seventh poem is called In the "Wilds Is a Dead Doe (*Ye Yu Si Kun*)" and contains direct descriptions and exclamations of the body's instinctive tendencies:

> "In the wilds there is a dead doe;
> With white rushes we cover her.

> There was a lady longing for the spring;
> A fair knight seduced her.
> In the wood there is a clump of oaks,
> And in the wilds a dead deer
> With white rushes well bound;
> There was a lady fair as jade.
> 'Heigh, not so hasty, not so rough;
> Heigh, do not touch my handkerchief-
> Take care, or the dog will bark'".

In "Song of Songs" (Song 7:12–13), the following line is given: "Bring me, O king, to your chambers. With you we rejoice and exult, we extol your love; it is beyond wine: how rightly you are loved!" (Song 1:4); furthermore, "He brings me into the banquet hall and his emblem over me is love" (Song 2:4). This is also one of the characteristics of love between men and women. We consider that love should be tangible and alive and that love between men and women often involves physical contact. Even if it has not yet occurred or is in the process of taking place, the differences and connections at the physical level profoundly influence the progression and manifestation of love between men and women.

The eighth poem is called "Peach-Tree (*T'ao Yao*)", which goes as follows:

> "Buxom is the peach-tree;
> How its flowers blaze!
> Our lady going home
> Brings good to family and house.
> Buxom is the peach-tree;
> How its fruit swells!
> Our lady going home
> Brings good to family and house.
> Buxom is the peach-tree;
> How thick its leaves!
> Our lady going home
> Brings good to the people of her house".

This poem describes the flourishing of flowers and leaves, symbolizing the sweet and beautiful stage of love. And a similar expression is found in the "Song of Songs", "The fig tree puts forth its figs, and the vines, in bloom, give forth fragrance" (Song 2:13). Therefore, we can see that the "Song of Songs" depicts the same wonderful season. Because of the great geographical differences, we cannot expect the Jerusalem area to have exactly the same vegetation as northern China. However, we find that both *Shijing* and the "Song of Songs" chose to describe a fruit-bearing tree and describe the beautiful state of these fruit trees during the fruit-bearing season. The difference is that in the "Song of Songs", the sentence describing the scent of flowers is placed after the fruit tree, which seems to imply that the scent of flowers is the more graceful level.

In summary, in comparing eight poems from *Shijing* with passages from the "Song of Songs", both texts reveal common portrayals of love, articulating a universal longing for idealized romance and depicting the simultaneous presence of joy and sorrow in relationships. Gift-giving among lovers, inherently spontaneous, underscores a mutual desire for connection without the expectation of return. Engaging in a romantic relationship amplifies the lovers' suffering due to temporary separations, fostering a yearning for continuous companionship. Physical intimacy serves as a marker of a deeper emotional bond, with both parties seeking greater fulfillment and happiness despite their physical limitations. Through the comparative analysis of *Shijing* and the "Song of Songs", identifying commonalities depicted across different civilizations, it can be posited that Chinese readers' engagement with *Shijing* would facilitate their understanding of Saint Bernard's sermons on the "Song of Songs".

### 3. Viewing *Shijing* through the Lens of Saint Bernard's Sermons

Saint Bernard devoted eighteen years (1135–1153) to the composition of his *Sermons on the Song of Songs*, placing profound emphasis on the experience of communion with the divine. He approached the "Song of Songs" with the spirit of a mystic, offering a mystical interpretation that underscored the intricacies of human and divine interaction. Regarding the comparison between Saint Bernard's *Sermons on the Song of Songs* and other vernacular literature, Brigitte Saouma (2016) and Marie-Madeleine Davy (1968) conducted some exceptional work. The innovation of this paper is the attempt to compare Saint Bernard's thoughts with *Shijing*. It is observed that his theological exegesis of the "Song of Songs", when applied to *Shijing*, reveals numerous ingenious parallels and congruences.

Before employing Saint Bernard's theology to analyze *Shijing*, the concept of "love" within Saint Bernard's theological thought is introduced first. It is imperative to acknowledge, at the outset, that an examination of Saint Bernard's writings might reveal an inclination towards a Christocentric interpretation of love. This paper does not purport to defend a doctrinal position. Nonetheless, a more nuanced comprehension of Saint Bernard's core perspectives is facilitated by an appreciation of the historical and personal context in which he operated, alongside the interpretive heritage of the "Song of Songs" within Christian theology. Saint Bernard's epoch was marked by the ascendancy of European Monasticism, distinguished by its advocacy for the eschewal of terrestrial pursuits in favor of celestial aspirations. Serving as an abbot, his literary corpus predominantly sought to explicate the Gospel and elucidate Christian doctrine. In the sphere of Christian exegesis of the "Song of Songs", prior to Saint Bernard, the commentaries by Origen of Alexandria and Saint Gregory the Great were notably influential. Their analyses converged on the allegorical identification of the Song's male protagonist with Christ and the female protagonist with either the Church or the individual soul. Saint Bernard endorsed this allegorical framework, with a particular emphasis on the latter allegory, thereby situating his interpretations within a well-established theological tradition.

Saint Bernard perceives love as an initial, albeit imperfect, state that, through deliberate effort, strives for improvement. This transformative process is inherently painful. He metaphorically relates the nascent monastic commitment to the evolving love between a man and a woman, highlighting the initial incompleteness of love (Bernard de Clairvaux 2006, p. 319). According to Saint Bernard, it is essential for an individual to continually engage in self-examination and purification to enhance their love for God. In Saint Bernard's view, as articulated by Christian Trottmann in *La Vision Béatifique*, this purification of love parallels a journey towards deeper understanding and assimilation, where purified love aligns more closely with divine charity and proximity to God (Trottmann 1995, pp. 99–100).

In the cosmic hierarchy, God reigns with ultimate majesty, distinguishing the Creator from the created humanity, with an infinite divide. "Her fervent love leads her to such intoxication that she disregards the Bridegroom's majesty" (Bernard de Clairvaux 2006, p. 161). Charity empowers humans to transcend fear, a fear that is rooted in the inherent unworthiness of divine love. Yet, through God's love, fear is cast out by perfect love. There is no fear in love, but perfect love drives out fear because fear has to do with punishment, and so one who fears is not yet perfect in love (1 John 4:18). It is noteworthy that the exchange of love between man and God is reciprocal, not transactional, predicated on genuine charity towards God (Bernard de Clairvaux 1993, pp. 102–4). This love, inherently rewarding, is not motivated by the expectation of reward, thus fostering a spontaneous and satisfying relationship with the divine.

In his first sermon of the *Sermons on the Song of Songs*, Saint Bernard posits that the "Song of Songs" occupies an exceptionally elevated status. Only a true lover will be capable of grasping its meaning. This canticle is not intended for the general populace; only those who sing and listen to it will have access. "It is truly a nuptial song, expressing the chaste and joyful embraces of the spirits, the harmony of customs" (Bernard de Clairvaux 1993, p. 76). For Saint Bernard, the relationship between humans and God is akin to that of a bride and bridegroom in a state of perfection.



Undoubtedly, Saint Bernard never read *Shijing*. Moreover, he knew no Hebrew, rendering him incapable of reading the original text of the "Song of Songs". If his interpretation of the "Song of Songs" was grounded in an understanding of human emotions and constructed using medieval philosophical and theological methodologies, then it should be permissible to apply his interpretative approach to *Shijing*. This endeavor is not aimed at doctrinal innovation but rather at exploring the similarities within human nature across different civilizations.

The opening of the "Song of Songs" states, "Let him kiss me with the kisses of his mouth! For your love is more delightful than wine. Your name is a perfume poured out; no wonder the maidens love you" (Song 1:2–3). Saint Bernard finds this opening quite abrupt, suggesting the presence of some unspoken matters preceding this initiation (Bernard de Clairvaux 2006, p. 66). In other words, there exists a prior offering of love, leading to the expressed desire for kisses in the opening of the "Song of Songs". "Similarly, the call in the poem *Piao Yu Mei* resonates with this sentiment: "seeking me are several gentlemen, may it come to its being lucky! seeking me are several gentlemen, may it come to (its being now) a decision now!", translating to an appeal for cherishing the moment and making a timely decision by the suitors. Evidently, the text of *Piao Yu Mei* also presupposes the offering of love by "several gentlemen". Saint Bernard interprets this as the divine's precedence in loving and seeking us before we do. This anticipation mirrors the "Patriarchs' fervent hope for the birth of Christ" (Bernard de Clairvaux 2006, p. 82). Through this interpretation, Saint Bernard elevates the simple mutual longing between a man and a woman to the collective human anticipation and hope for the Savior, namely, Christ.

In his third sermon (Bernard de Clairvaux 2006, p. 108), Saint Bernard offers an interpretation of "Song of Songs" 1:7, which goes, "Tell me, you whom my heart loves, where you pasture your flock, where you give them rest at midday, lest I be found wandering after the flocks of your companions". He posits that this verse contrasts two states of being. The first state is one in which a person cannot find his beloved and thus fails to approach her, "lying in ashes" (Bernard de Clairvaux 2006, p. 108), experiencing pain and disgrace. In this state, the lover's yearning to find the beloved echoes the cry in "Come, says my heart, seek his face; your face, Lord, do I seek!" (Psalms 27:8) and in "Tell me, you whom my heart loves, where you pasture your flock, where you give them rest at midday, lest I be found wandering after the flocks of your companions" (Song 1:7). This is akin to the sentiments expressed in the poem *Kiao T'ung* of *Shijing*, "it is all your fault, but it makes me unable to eat" and "it is all your fault, but it makes me unable to rest". Saint Bernard suggests that the second state is described by "In the morning let me hear of your mercy, for in you I trust. Show me the path I should walk, for I entrust my life to you" (Psalms 143:8), "where at sunrise, the soul of the servant is gladdened" (Psalms 86:4), and "abounding joy in your presence" (Psalms 16:11). Unlike the first state, this second state is characterized by an interesting point Saint Bernard makes is the dual nature of the beloved as both sweet and stern, "Ah! Sweet Lord, Ah! Severe Lord!" (Bernard de Clairvaux 2006, p. 110). The sweetness stems from His goodness and beauty, while the severity comes from His absence, thereby creating a tension that is the source of the seeker's pain. The direct statement from *Kiao T'ung*, "it makes me unable to rest", encapsulates this pain of seeking. This represents the essence of love and the reality of life on earth. Even when one finds God and enters into a loving relationship with Him, the sufferings of the world are unavoidable, and love cannot be wholly sweet. Saint Bernard argues that the fluctuation between joy and pain in the love described in the "Song of Songs" is due to the incompleteness of love, not a flaw in love itself. Similarly, the portrayal of the beloved in *Kiao T'ung* reflects this mixture of pain and joy, not because the relationship is malevolent but because love has not reached its fulfillment. According to the interpretation of Saint Bernard, this condition, referenced in both texts, is a manifestation of human common sense. Experiencing pain in love does not imply that love is fundamentally flawed. Instead, it indicates that love, in its nascent stages and rooted in the natural realm, may encompass suffering, risk, and potential self-harm. Saint Bernard argues that love, which he believes is

derived from a transcendent source—God—necessitates external moderation at this stage. As a consequence of such transcendental moderation, love becomes more beautiful and perfected over time (Bernard de Clairvaux 1993, pp. 118–20).

The exchange of gifts between lovers is a commonplace occurrence. Saint Bernard's focus is not on the symbolic significance of the items designed as gifts, which often varies greatly across different cultures. Rather, Saint Bernard uncovers an intriguing paradox in this matter; he proposes that the glory of love is only attainable by the recipient when they believe themselves to be unworthy of such a gift. This is because if the recipient deems himself deserving of beautiful gifts, this implies a sense of obligation on the part of the giver. Consequently, upon receiving a gift, the recipient would not experience surprise, and thus, the glory and joy of love remain unfelt. However, when the recipient considers himself unworthy of beautiful gifts and receives them nonetheless, they realize that despite their unworthiness, the lover compensates this deficit with love, bestowing upon the recipient the symbolic glory of the gift and, therefore, happiness (Bernard de Clairvaux 2000, pp. 220–22). Saint Bernard evidently approaches the concept of "a lover's gift" with certain presuppositions. His premise is that the lover represents the supreme God, with a significant status disparity between the lover and the beloved. True love, however, transcends this distance, be it physical or psychological. More importantly, in the realm of love, the act of giving gifts occurs naturally. The verses of *Tsing Nü* do not explicitly involve a deity of immense dignity, yet in human romantic relationships, not all are characterized by complete equality in social status. The authors of *Shijing* must have observed this as well, suggesting that, even though they might not have contemplated a divine being with personhood, the depiction of romantic relationships in *Tsing Nü* can fully accommodate the image of God and the relationship of divine love as described in Saint Bernard's theology. In the act of gift-giving between lovers, Saint Bernard discerns underlying issues, including whether the glory represented by the gift is deserved by the recipient, and how to confront disparities in status among lovers. Within the scenarios depicted in the "Song of Songs", Saint Bernard advances the notion that pure love enables lovers to overlook differences in status (Bernard de Clairvaux 2006, p. 163), and this is certainly true in *Shijing*. Discrepancies in social standing, and even ontological gaps, do not hinder the act of loving between individuals. Hence, the purer the love, the more capable it is of penetrating distances.

In his twenty-fifth sermon, Saint Bernard offers commentary on "Song of Songs" 2:10, and his insight can be very useful for a comparative analysis of the below two texts, "even though I have not gone (to you), why do you not come?" in *Tsi Kin*, and "Arise, my beloved, my beautiful one, and come" in Song 2:10. Both texts evidently express a longing for the beloved's arrival. However, Saint Bernard interprets this longing from a more symbolic perspective, situating this anticipation within the framework of chronology, where "the arrival of the beloved" has not yet occurred. Undoubtedly, this event is characterized as "imminent" or "in the future", for if there is no willingness to "arrive" on the part of the anticipated one, it would not constitute a relationship of love. Saint Bernard's focus is on the aspect that "the lover is about to arrive but has not yet arrived" (Bernard de Clairvaux 1998, p. 262). He suggests that this signifies the incomplete journey towards Christ, meaning one is progressing on the correct path but has not yet reached the destination. This interpretation also applies to the line "I am dark but lovely" (Song 1:5). The apparent contradiction of being "dark but lovely" mirrors the actual condition of those enmeshed in worldly love. Lovers of the world are imperfect, and despite feelings of unworthiness and inferiority, love endows individuals with the strength and courage to pursue. Utilizing Saint Bernard's interpretation to reconsider the poem *Tsi Kin*, one might propose the following interpretation: if "you" (*Tsi*) refers to the Savior, Christ, then the "yet to come" of "you" (*Tsi*) can be understood as the yearning and anticipation for the Savior by those who have not yet heard the Gospel. The anticipation of a beloved's presence constitutes a quintessential human sentiment. Yet, from an analytical perspective, the existential condition of humanity predicates that the beloved can never

fully manifest in the temporal realm. This is attributed to the fact that, notwithstanding the physical proximity of the beloved, an everlasting cohabitation (both in physical and spiritual terms) remains unfeasible. Saint Bernard posits that this phenomenon underscores the intrinsic imperfection of terrestrial love. However, individuals who have savored the sweetness of affection are predisposed to endure contemporary tribulations in the pursuit of transcendent love. This alludes to the celestial communion between humans and the divine in the afterlife (Bernard de Clairvaux 1993, p. 130).

Lovers, when yearning for their beloved, invariably experience a mix of joy and distress. Saint Bernard de Clairvaux (1998, p. 222) also notes the scenario where the lover lies in bed, unable to sleep. He considers this as a paradoxical state. Initially, lying in bed attempting to sleep ostensibly reflects a sense of security, a state of tranquility. This represents the lover's sense of safety within the romantic relationship. However, the person lying in bed does not truly find peace, as they are unable to fall asleep, anxiously pondering over their romantic ties. Saint Bernard terms it "the sweet sorrow" (Bernard de Clairvaux 1998, p. 222). As implied in *Kwan-Ts'ü*, this sorrow is characteristic of an early stage in love, where mutual understanding and trust are less established. Within the lexicon of his medieval mystical theology, Saint Bernard describes, "This is not the room of the husband" (Bernard de Clairvaux 1998, p. 224). The essence of love is such that one feels restless upon attainment yet yearns for reacquisition upon loss. It is precisely because "This is not the room of the husband" that the lover cannot attain perfect rest. In other words, only if the lovers perfectly unite, entering a state of perfection, can this restlessness be alleviated. Viewing this sentiment from the perspective of the lover, we find it to be both joyful and sorrowful. However, this does not imply that the relationship must progress to perfection. Further analyzing the scenario described jointly by *Kwan-Ts'ü* and "Song of Songs" 5:2, we ascertain that the root of this distress is fundamentally the absence of the beloved. As not every love story culminates in fulfillment, the absence of the beloved does not necessarily mean he or she will be present in the future. Building on the scenario depicted in *Kwan-Ts'ü*, we can envisage the subsequent journey of the couple, potentially culminating in perfect union or separation. In his fifty-sixth sermon, Saint Bernard analyzes two outcomes and directions for the absence of the beloved (Bernard de Clairvaux 2003, pp. 142–44). The first scenario involves the beloved's temporary departure out of compassion, posing no significant harm, whereas, in the second scenario, the beloved departs in anger, signifying the failure of love. The former is due to the lover being a holy person, while the latter is because the lover is sinful and unwilling to repent. In "Song of Songs" 5:2, which states, "my sister, my beloved, my dove, my perfect one!" Saint Bernard sees this as beautiful imagery, symbolizing the absence of terror in love (Bernard de Clairvaux 2003, p. 135). Notably, the trees and rivers described in *Kwan-Ts'ü* are also favorable symbols. According to Saint Bernard's mystical theology, there is no fear in love. Even in its nascent stages, love is not an entity of ugliness and evil. Even though lovers may experience suffering, it does not stem from inherent malice or wicked intentions. Being referred to as a "dove" highlights the intimacy and trust within love.

In his sixth sermon, Saint Bernard quotes a woman's grand praise of her husband's legs, which are likened, according to the woman, to be "columns of marble resting on golden bases" (Song 5:15). In his view, the praise of beauty is essentially a celebration of virtue, with the most perfect virtues embodied in Jesus Christ. Therefore, we can deduce that, following Saint Bernard's interpretative approach, the laudation of the lover's physical beauty symbolizes the love for Christ (Bernard de Clairvaux 2006, p. 148). In his twenty-eighth sermon, Saint Bernard analyzes the lover's ruddy cheeks and beauty. According to St. Bernard, the ruddy hue of the cheeks serves as a symbol of martyrdom, while beauty symbolizes virginity (Bernard de Clairvaux 1998, p. 366). Martyrdom signifies the lover's sacrifice for the beloved, even to the point of offering his own life; virginity signifies the lover's esteem for the beloved, hence the willingness to endure and wait. From a natural perspective, Saint Bernard emphasizes the significance of "ruddiness" and "beauty" as indispensable in a loving relationship. This is also reflected in *Kien Hi*, with phrases like

"he is shining as if smeared with red" and "The tall man is very great". Furthermore, Saint Bernard references other biblical passages: "Who is this that comes from Edom, in crimsoned garments, from Bozrah—This one arrayed in majesty, marching in the greatness of his strength?" (Isaiah 63:1), and "crowned him with glory and honor" (Psalms 8:6). He situates the romantic sentiments described in the "Song of Songs" within a broader narrative of salvation, viewing them as symbols of the lover and as manifestations of perfect love with Christ. In worldly romantic relationships, one naturally praises the physical and facial beauty of the beloved, even when such beauty may not be widely acknowledged. However, out of love, one naturally perceives it as beautiful. Saint Bernard introduces a symbolic meaning, interpreting the praise of beauty as a willingness to sacrifice and to show esteem, which is an aspect often valued in romantic relationships. In essence, according to Saint Bernard, there is no absolute disconnection between worldly love and love for Christ; noble qualities in earthly love can also symbolize aspects of the relationship between humans and Christ within the history of salvation.

In his twenty-third sermon, Saint Bernard comments on the depiction of a more intimate relationship between lovers as described in the "Song of Songs", which is also portrayed in *Ye Yu Si Kun*. Saint Bernard notes that such an intimacy initially arises from an attraction to the beloved's virtuous characteristics, metaphorically referred to as "perfume" in the "Song of Songs" (Bernard de Clairvaux 1998, p. 198). Following the experience of deeper intimacy, the relationship between the lovers advances further. Saint Bernard describes this as "entering the bedroom". Entering the bedroom symbolizes a state of consummation in the love relationship. Here, Saint Bernard references Saint Paul's description, "We no longer see God there as troubled with anger or held back by his occupations, but may discern what is the will of God, what is good and pleasing and perfect" (Romans 12:2). This vision does not frighten, but rather enchants. Indeed, "it doesn't arouse anxious curiosity, on the contrary, it calms it; it doesn't tire the senses but reassures them. This is true rest. The God of serenity makes all things serene. To contemplate Him in His rest is to rest oneself" (Bernard de Clairvaux 1998, p. 234). In Saint Bernard's view, intimacy no longer pertains to the initial unease and trepidation of love but moves toward peace and fulfillment. Individuals find rest and rejuvenation in the tranquility of beautiful love. In his seventh sermon, Saint Bernard emphasizes that the depiction in "Song of Songs" 2:4 is not a celebration of carnal desires. "Indeed, her love is chaste, for she seeks the one she loves and nothing that belongs to him. Her love is holy, for she loves not in the lust of the flesh but in the purity of spirit. Her love is fervent, for she is so intoxicated with this love that she no longer thinks of the majesty of the Husband" (Bernard de Clairvaux 2006, p. 158). According to Saint Bernard, the intimacy of love includes the physical aspect but is not limited to it. The spirit of love transcends the natural order, allowing the heart to approach God's altar, thus highlighting the transcendence of love. Evidently, Saint Bernard's interpretation extends beyond the literal meaning contained within the texts of the "Song of Songs", yet it is not contrary to the spirit contained within these texts. Though it is not a religious text, one can certainly apply the same principle of interpretation to *Ye Yu Si Kun*. Through simple human experiences, we find this also to be an expression of perfect love across various cultures. Within the context of Christianity, it is a manifestation of the relationship of love between God and humanity.

"Song of Songs" 2:13 and *T'ao Yao* exhibit a certain similarity. Although peach and fig trees are distinct species, these plants can be considered interchangeable from the perspective of symbols of love. Saint Bernard posits that the sweetest time in the air is when the fig tree produces its green fruit (Bernard de Clairvaux 2003, p. 222). The fig tree serves merely as a symbol, representing human love. Despite being unripe, naive, and limited, it still embodies the potential for infinite goodness. Saint Bernard suggests that fruit-bearing trees resemble individuals whose behavior is more "friendly". The lush foliage and abundant fruit are the result of the tree's collective effort, embodying a spirit of union and communion, both of which are vital and fruitful. From a natural standpoint, the fruit tree is already perfect (Bernard de Clairvaux 2000, pp. 236–38). As stated in "Song

of Songs" 2:13, the fruit tree merely represents a preparatory stage. The fruit of the tree may not always be sweet, but the fragrance is invariably delightful. Fruit symbolizes the initial stage of love, while fragrance represents the fulfillment of love. Therefore, superior is the fragrance of grapes. Saint Bernard does not dismiss natural love; he merely considers it insufficiently beautiful. With the addition of grace, the bitterness of love is removed. He praises the love among lovers in the world yet considers it an imperfect manifestation of the love between people and God. External beauty is imperfect, whereas the scent of flowers, though invisible, symbolizes perfect love. Quoting Paul's famous maxim, "Love never fails. If there are prophecies, they will be brought to nothing; if tongues, they will cease; if knowledge, it will be brought to nothing" (1 Corinthians 13:8). Saint Bernard explains that in the perfect stage of love, many external manifestations disappear, and he makes an analogy to grape vines. Saint Bernard deems this love of vines as loving God "with all your heart, with all your soul, and with all your strength" (Bernard de Clairvaux 2003, p. 238). It progresses from the stage of the fig fruit towards perfection. The process requires acts of charity, repentance, and letting perfect love cast out fear. The former seems more pronounced in love for one's neighbor, while the latter is in love for God. According to Saint Bernard, these are not in opposition or fundamentally different but vary according to the stages of love. If we liken the peach trees described in *T'ao Yao* to the fig trees in the "Song of Songs", we find that the sweet love described by the peach trees implies the potential to become more perfect. The beauty described by peach trees has a very limited temporal boundary. The lady described in *T'ao Yao*, although in a state of beautiful happiness, undoubtedly needs an open mind towards an eternal world if she seeks longer-lasting happiness. This is precisely the theological perspective that Saint Bernard constructs from his analysis of "Song of Songs" 2:13.

## 4. Conclusions

Through the comparison and analysis of the *Shijing* and the "Song of Songs", similarities have been identified that symbolize the universal aspects of human nature in the perception and expression of love. These parallels blaze a new trail, suggesting the possibility for Chinese readers to embrace Saint Bernard's theological interpretations of the Songs of Songs. The potential for such cross-cultural understanding is not only viable but also enriches Chinese readers' comprehension of medieval theology in a more holistic manner. Saint Bernard authored numerous works, among which the *Sermons on the Song of Songs* is a significant component. By leveraging their understanding of the *Shijing*, Chinese readers can more effectively engage with Saint Bernard's theology, thereby circumventing biased interpretations of his writings. Nygren (1953, p. 649) sharply differentiates between *eros* and *agape*, positing that Saint Bernard's conception of love is predominantly *eros* with Neoplatonic influences. Nygren critiques Saint Bernard and his medieval contemporaries for not attempting to reformulate the Caritas synthesis, instead persisting in an *eros*-based understanding of love. He likens Saint Bernard's theoretical framework of love to a Gothic church: secularly grounded yet striving heavenward. Despite their aesthetic appeal, Nygren contends that such structures symbolize the theoretical foundation of love as "something as earthly and human, far too human, as natural self-love" (Nygren 1953, p. 650), failing to reach the divine. He suggests that Saint Bernard's sophisticated treatment of Caritas inadvertently results in a moralistic love doctrine "as remote as possible from the Agape-love of Christianity" (Nygren 1953, p. 651). In my view, Nygren's interpretation of Saint Bernard is biased. Nicolas Perrier critiques Nygren for exclusively concentrating on *On Loving God*, neglecting Saint Bernard's *Sermon on the Song of Songs* (Perrier 1953). Nygren adopts a binary viewpoint, whereas Saint Bernard acknowledges the ambiguity of affections, advocating for a singular love that necessitates purification—a goal of human pursuit, unattainable in the earthly domain.

Dom Fernand Cabrol OSB contends that Saint Bernard exhibits a lack of humanitarian concern, analyzing the divergent spiritual orientations of Cluny and Cîteaux, with the latter centered around Saint Bernard (Cabrol 1928). He identifies two distinct thoughts

since the inception of Christianity: the asceticism represented by John the Baptist and the humanitarianism of Jesus Christ. While acknowledging Saint Bernard's simplicity, passionate love for God, and self-sacrifice, Cabrol overlooks his interpretation of the "Song of Songs". In Saint Bernard's writings, an appreciation for worldly aspects is evident. To him, the love between a man and a woman is not only beautiful but also crucial, mirroring the perfect relationship between humans and God. The relationships among people, including those with neighbors and guests, are significant as they embody the practice of charity.

Ironically, Nygren accuses Saint Bernard of prioritizing human nature over the divine, favoring human love over sacred love, while Cabrol charges him with an overemphasis on asceticism, neglecting human aspects. These disparate critiques of Saint Bernard intriguingly unifies in their collective misreading of his interpretation of the "Song of Songs". The notable misinterpretations of Saint Bernard by Nygren and Cabrol fundamentally stem from their neglect of his exegesis on the "Song of Songs".

This paper reveals substantial similarities between *Shijing* and the "Song of Songs" in their representations of love. Understanding Saint Bernard's mystical theology through these similarities, Chinese cultural readers can sidestep the errors Nygren and Cabrol encountered.

Is there someone who loved me before I expressed my love? Pursuing this query to its ultimate conclusion, what do we discover? In love, the intertwining of suffering and joy prompts the following question: how can one achieve purely joyful love? In addressing love's inherent inequalities, what approach should lovers take? Given that profound love often seems transient, can eternal love exist? Approaching these concerns shared by *Shijing* and the "Song of Songs", in an attempt to comprehend Saint Bernard's mystical theology, we may arrive at a more comprehensive doctrine of love as proposed by Saint Bernard. It is neither purely secular, carnal, worldly love nor entirely ascetic or self-denying love. Rather, it is a form of affection that can manifest between men and women but is not confined to this; it is purified continuously in the world and put into practice through actions, thereby gradually approaching divine love.

Discussions on love remain perpetually captivating and inexhaustible, with romantic love between men and women representing one of the most intimate and enigmatic forms among various types of love. It demands the holistic engagement of individuals, encompassing both the body and the soul. The "Song of Songs" reflects the divine communication to humanity, while *Shijing* embodies the early philosophical thought in Chinese civilization and is among the few Confucian texts that directly depict romantic love. Through the analyses of this paper, we discover significant parallels in the early love poetry within both Chinese and Hebrew civilizations, indicating their openness to transcendence. Although Christianity does not regard *Shijing* as divine revelation but something that originated from humanity's most fundamental and simple emotions, an understanding of the love described in *Shijing* can aid readers within Chinese culture to comprehend the "Song of Songs" and enter into the context of Saint Bernard's mystical theology. Saint Bernard identified a pathway between human love and divine love. For readers within Chinese culture, *Shijing* serves as a steppingstone onto this path.

**Funding:** This research received no external funding.

**Institutional Review Board Statement:** Not applicable.

**Informed Consent Statement:** Not applicable.

**Data Availability Statement:** No new data were created or analyzed in this study. Data sharing is not applicable to this article.

**Conflicts of Interest:** The author declares no conflict of interest.

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
