# Peer review of "Approaching Saint Bernard’s Sermons on the “Song of Songs” through the Book of Odes (Shijing): A Confluence of Medieval Theology and Chinese Culture"

_religions, doi:10.3390/rel15040513_

Round 1
Reviewer 1 Report
Comments and Suggestions for Authors
The manuscript on "Approaching Saint Bernard's Sermons on the Song of Songs through the Book of Odes (Shijing)" is a wonderful attempt to address a comparison that is obviously very important and potentially interesting, but hasn't really been worked through by anybody to my knowledge. The author appears to have a profound knowledge of Saint Bernard's writings, though perhaps only a more cursory understanding of the Chinese poems, but it would be hard for anybody to master both discourses from such different domains. Anyway, as the author correctly argues, there is a compelling comparison to be made here between the depictions of love in these two bodies of poetry, as well as the Christian moralization and theological interpretation of the Book of Odes.
Though this is already a promising paper, it could be improved substantially in two ways: by addressing the problem of the allegorization of the Book of Odes; and by using other scholarly sources to present those poems.
1) In the attempt to sketch a comparative "confluence," one key element is missing. We have Song of Songs -> Bernard's interpretation in parallel to Book of Odes -> whose interpretation???. In other words, just as there is a long and rich tradition of interpreting the Song of Songs, beginning with Judaic theology itself, there is also a two-thousand year tradition of interpreting the love poems in the Book of Odes in various ways. By failing to mention this tradition, the author gives a skewed picture of the Chinese side of the confluence. In reality, the meaning of the Book of Odes poems is not at all transparent, and Chinese scholars have long struggled to provide moralizing readings of the poems in harmony with Confucian thought. One source that could be cited here is Steven van Zoeren, Poetry and Personality, or perhaps better Haun Saussy, The Problem of a Chinese Aesthetic. The author doesn't need to resolve these issues but ought to give a sketch of the dynamic. (Essentially, the poems have always been understood to have political and moral implications, whether or not they are love poems, and their Chinese interpreters have struggled to resolve difficult questions about aesthetic and ethics in the field of their interpretations.)
2) Karlgren is not the most approachable or convenient source for the purpose of this comparison. Ideally, of course, one would consult modern Chinese scholarship and other sources. If the author is not competent in Chinese, there are still better sources to be used. I would recommend the Joseph Allen updating of Waley's Book of Songs, published in 1996, for the main translations and titles of the poems. This is conveniently arranged in the traditional numbering. But to get a sense of the Confucian interpretations that have dominated discourse on the Book of Odes, I would consult James Legge's translation The She King. In his footnotes Legge regularly summarizes the views of various premodern Chinese scholars, including the "Minor Prefaces" which introduce each of the poems. For instance, summing up "Jian xi" as a poem "which praises the body of the beloved one" is not really accurate at all, as this is a poem about the performance of a particular dance.
My two suggestions would involve a substantial amount of work, but I feel it would be well worth it, because this such a promising line of inquiry, and I would myself look forward to seeing the author's results.
Comments on the Quality of English LanguageEnglish is mostly good but requires more editing. "shared oversight" and "renowned misinterpretations" are both wrong, on p. 12.
The introductions and summaries of the Chinese poems are unfortunately quite sloppily done, e.g. "The first poem.." on p. 3. It is not the first poem in the anthology but seems to have been chosen at random, and needs to be introduced properly. Nor are these poems "typical" of the Book of Odes. On p. 4, " a beloved classic that remains popular and relevant in today's Chinese society" requires supporting evidence. Similarly, "The sixth poem is so classic..." on p. 5 makes no sense.
All the poems should be presented as block quotes of poetry with line breaks, not as prose.
Author Response
First and foremost, I extend my deepest respect and heartfelt gratitude towards your exceptional professionalism and unwavering commitment as a reviewer. Your insightful suggestions have been of paramount importance and remarkably pertinent to the refinement of this manuscript.
Heeding your invaluable advice, I delved into the works of Steven van Zoeren and Haun Saussy, which proved to be significantly enlightening for my manuscript. I acknowledge that my initial overlook of the rich interpretative tradition surrounding the Shijing was a simplification that did not do justice to its complexity. Inspired by the thoughts of these esteemed authors, I have undertaken revisions to enhance my manuscript accordingly.
Concerning the selection of translation versions, I concur that each exhibits unique traits, highlighting the inherent challenges and beauty of cross-cultural and linguistic exchanges. In alignment with your recommendations, my primary reference was the version found in Waley's Book of Songs (1996), a choice inspired by its elegance and precision. Nonetheless, I have preserved Karlgren's translations of the poem titles, valuing the phonological insights they offer as integral to understanding the Shijing's essence. Furthermore, I consulted a French translation by Pierre Vinclair (2019), uncovering nuances not previously accentuated in English translations. This comprehensive approach has substantially addressed the concern you raised about the absence of succinct introductions to each poem, with the refined translations offering clearer and more aesthetically pleasing renditions. I have also incorporated my interpretations for certain poems, such as "Jian Xi (So Grand)", which I believe further elucidates their meanings.
As for the organization of poems within the manuscript, I have elaborated on my rationale, earnestly hoping for your understanding and acceptance of my perspective.
Your guidance has been instrumental in this endeavor, and I am immensely grateful for the opportunity to benefit from your expertise.
Reviewer 2 Report
Comments and Suggestions for Authors
As I understand it, this project engages a comparative study of the Shijing and the works of St. Bernard. It does so with the expressed intent of showing how the two share expressions of love, a sentiment that transcends cultural and historical boundaries. On the surface, I guess this is a viable project (that said, I’m always interested in the methodology of such comparisons, that is, why these two pieces of literature and not others?). This paper does not, however, live up to its intended goals.
Most of the paper is a treatment of St. Bernard’s work. True, you address several of the poems in the Shijing, but most of this treatment is amateurish and impressionistic. It’s almost as if you assume a Chinese audience that is intimately familiar with this material so that you do not carry the burden of exposition. This is a mistake. You should assume your audience is not so familiar with the Chinese material (unless this is a volume solely intended for Chinese audience). To be sure, some of the comparisons are forced and unpersuasive. What is more, the piece comes off as a bit Christocentric. Do you really want to suggest – see lines 432-435 – that the “you” in the Chinese poem is alluding to the Savior? This seems wildly unsubstantiated, if not in fact impossible. Also, be careful that you do not endorse too strongly heteronormativity… do you want to suggest that relationships other than those between a man and a woman fail to achieve the same type of “sacrality”? I found the discussion of Bernard’s critics at the end of the paper out of place and therefore a bit unhelpful. If you want to contextualize Bernard’s bridal mysticism by introducing its critics, then do so earlier in the paper and allow the rest of the paper to answer the criticism, should that be your desire. Similarly, the introduction of Gadamer’s fusion of horizons is gratuitous, insofar as that material doesn’t factor into the larger discussion. I wonder if the same cannot be said for your allusion to Marion…. However, on the latter point, I believe some interesting work can be done.
You seem to suggest that there is a difference between transactional love and “true” love. This holds out some promise. You suggest that there may be an ontological gap… I believe you could develop this. But, to do this will require an engagement with Husserl’s transcendental phenomenology. On that note, there are plenty of authors, Marion included, who have taken up the task of an ethical criticism of Husserl. You should probably consider some of Derrida’s work on the messianic and Levinas’s work on ethics as first philosophy. One can argue, for instance, that the transcendental absence of the other is the true condition for selfless love. Similarly, you may want to distinguish, say, eros from agape. You may also want to consider – perhaps for further work in this area – that the Hindu tradition draws distinctions between kama (sexual love) and prema (divine love). What is more, there is a Hindu theory of viraha bhakti wherein the loved other is always constituted by an ontological gap in the subject’s intentional horizon, resulting in a love for the withdrawn other. Lastly, perhaps some of the Daoist material could be interesting. The Dao De Jing suggests that there is a complement between substance and absence/nothingness (just a quick thought).
I like projects in comparative philosophy, religion, and literature. This project falls within that purview. However, as it stands, this piece needs some substantial revision before it should go to press.
Comments on the Quality of English LanguageThe quality of English language is decent. There are some places for improvement.
Author Response
Firstly, allow me to extend my sincerest gratitude for the exceptional professionalism and expansive academic insight you have brought as a reviewer to my manuscript. Your dedication to this endeavor commands the highest respect, and as a doctoral candidate, I am profoundly touched and inspired by your guidance.
The selection of the two texts under discussion in this manuscript stems from a blend of personal interest and the significant cultural and thematic relevance each holds, as detailed in the manuscript. The overarching goal of this paper, as you astutely observed, is to bridge a cross-cultural understanding by offering readers within the Chinese cultural context a gateway into the mystical theology of Saint Bernard, aligning with the thematic essence of this special issue. The mysticism of medieval Europe represents an array of concepts quite alien to a Chinese audience. My endeavor seeks to unearth parallel elements within the Shijing and the Song of Songs, laying a foundational stone for this ambitious cross-cultural dialogue. It is my sincere hope that you find this approach and rationale acceptable.
In response to your invaluable feedback regarding the initial treatment of the poems as overly simplistic, I have adopted a more nuanced and detailed English translation, supplemented by insights from a French version, and enriched the manuscript with additional interpretations. My aim is to more effectively capture and convey the profound emotions embedded within the Shijing, enhancing the reader's comprehension and appreciation.
I am particularly thankful for your forthright identification of the paper's inadvertent Christocentric portrayal. This observation is crucial, as it was never the intention of this study to adopt an apologetic stance. Given Saint Bernard's significant historical and religious context, the Christocentric nature of his writings is undeniable. I have, therefore, incorporated clarifications within the manuscript to carefully distinguish between Saint Bernard's perspectives and the broader objectives of this study. Regarding the interpretation of "the ‘you’ in the Chinese poem as alluding to the Savior," I concur with your assessment. Such an interpretation strays from the intention of the Chinese poems and indeed, the identity of the savior is not the focal point of this study. Instead, I explore the shared human sentiment of affection, intrinsic to both texts, as a testament to the universality of human experience.
Your critique concerning the paper's handling of heteronormativity was rightly pointed out, and I have addressed this significant oversight in the concluding sections of the manuscript, ensuring a more inclusive representation of affection. The paper now explicitly acknowledges that "it is a form of affection that can manifest between men and women but is not confined to this."
The inclusion of a critique of Saint Bernard in the conclusion was indeed an unconventional choice. My intention was to introduce a dimension of intellectual curiosity and engagement, encouraging readers to reflect on the broader implications of the viewpoints presented.
Your profound expertise in phenomenology is both admirable and inspiring. While the limited scope and focus of this manuscript preclude an extensive discussion on phenomenology, the addition of perspectives from Levinas has been made in acknowledgment of your insightful suggestions. Your recommendations have been enlightening and will undoubtedly shape the direction of my future research endeavors.
The rich historical and cultural tapestry of Hinduism is an area I regret not exploring further in this manuscript, constrained by linguistic and resource limitations. Nevertheless, your encouragement has broadened my academic horizon, illuminating the potential for similar explorations across varied cultural landscapes.